# Methodological Considerations for Neuroimaging in Deep Brain Stimulation of the Subthalamic Nucleus in Parkinson’s Disease Patients

**DOI:** 10.3390/jcm9103124

**Published:** 2020-09-27

**Authors:** Bethany R. Isaacs, Max C. Keuken, Anneke Alkemade, Yasin Temel, Pierre-Louis Bazin, Birte U. Forstmann

**Affiliations:** 1Integrative Model-based Cognitive Neuroscience Research Unit, University of Amsterdam, 1018 WS Amsterdam, The Netherlands; jmalkemade@gmail.com (A.A.); pilou.bazin@uva.nl (P.-L.B.); buforstmann@gmail.com (B.U.F.); 2Department of Experimental Neurosurgery, Maastricht University Medical Center, 6202 AZ Maastricht, The Netherlands; y.temel@mumc.nl; 3Municipality of Amsterdam, Services & Data, Cluster Social, 1000 AE Amsterdam, The Netherlands; mckeuken@gmail.com; 4Max Planck Institute for Human Cognitive and Brain Sciences, D-04103 Leipzig, Germany

**Keywords:** Parkinson’s disease, magnetic resonance imaging, deep brain stimulation, ultra-high field

## Abstract

Deep brain stimulation (DBS) of the subthalamic nucleus is a neurosurgical intervention for Parkinson’s disease patients who no longer appropriately respond to drug treatments. A small fraction of patients will fail to respond to DBS, develop psychiatric and cognitive side-effects, or incur surgery-related complications such as infections and hemorrhagic events. In these cases, DBS may require recalibration, reimplantation, or removal. These negative responses to treatment can partly be attributed to suboptimal pre-operative planning procedures via direct targeting through low-field and low-resolution magnetic resonance imaging (MRI). One solution for increasing the success and efficacy of DBS is to optimize preoperative planning procedures via sophisticated neuroimaging techniques such as high-resolution MRI and higher field strengths to improve visualization of DBS targets and vasculature. We discuss targeting approaches, MRI acquisition, parameters, and post-acquisition analyses. Additionally, we highlight a number of approaches including the use of ultra-high field (UHF) MRI to overcome limitations of standard settings. There is a trade-off between spatial resolution, motion artifacts, and acquisition time, which could potentially be dissolved through the use of UHF-MRI. Image registration, correction, and post-processing techniques may require combined expertise of traditional radiologists, clinicians, and fundamental researchers. The optimization of pre-operative planning with MRI can therefore be best achieved through direct collaboration between researchers and clinicians.

## 1. Introduction

Longevity is increasing and consequently triggering a surge in age-related, multimorbid neurodegenerative diseases [1,2]. One of these diseases is Parkinson’s disease (PD). PD is the second most common neurodegenerative disorder worldwide and typically occurs after 50 years of age [3]. This is a multi-systems disease primarily characterized by symptoms that affect movement control, such as bradykinesia, tremor, rigidity, postural instability, and gait difficulties [3].

Drug treatments for PD are symptomatic in nature and function to replace the dopamine deficiency within the brain that occurs due to loss of nigrostriatal dopamine neurons [4,5,6]. While dopaminergic medications relieve the motor-related symptoms of PD, they do not address non-motor symptoms, further complications, or disease progression [6]. Moreover, drug therapy in PD is associated with side effects that include but are not limited to nausea and vomiting, sleep disorders, hallucinations, and delusions. Furthermore, as the disease progresses, initially beneficial drug treatments become less effective in about 40% of patients. At this stage, the therapeutic window begins to narrow and the medication may wear off faster, resulting in the re-emergence or worsening of motor fluctuations [7,8]. Chronic drug treatment and disease progression are also associated with levodopa-induced dyskinesias, which refer to involuntary, uncontrolled movements that occur when medications are most effective [7,8,9]. Increasing the dosages in response to reduced durability of levodopa or dopamine agonists is not always feasible. Alternative treatments such as device-aided therapies may then be considered.

The next step for a subset of patients is neurosurgery intervention by means of deep brain stimulation (DBS) of the subthalamic nucleus (STN) [10,11,12,13]. The STN is a small, glutamatergic, biconvex structure with a high iron content that is located within the subcortex [14,15]. DBS involves the implantation of electrodes that emit persistent high frequency stimulation in this nucleus [11,12,13]. The STN is a viable target for DBS as it modulates output of both the indirect and hyper-direct cortico-basal pathways, whose functions are assumed to suppress undesirable motor behavior and inappropriate movements, respectively [16,17]. In PD, dopaminergic degradation of the substantia nigra (SN) is thought to result in inhibition of direct pathways, as well as disinhibition of indirect and hyper-direct pathways. Collectively, this leads to the functional disinhibition of output to motor-related areas of the cortex, which is thought to produce impaired movement and reduced movement control [16]. However the exact mechanisms underlying DBS are still poorly understood, although the general consensus is that DBS results in a functional normalization of pathologically overactive circuits [17,18,19].

While DBS may ameliorate between 60 to 90% of the motor-related symptoms of PD, it can produce neuropsychiatric side effects and emotional or associative disturbances, with side effects ranging from hypomania; apathy; hallucinations; and, as well as general changes in moral competency, personality and reckless behavior [20,21,22,23]. A fraction of patients will fail to exhibit a long-term clinical benefit in the reduction of parkinsonian symptoms [24,25]. Revisions or removals of the DBS system occur in between 15 and 34% of operated patients, 17% of which are attributed solely to electrode misplacement [26,27]. Additional risks can arise from the surgery itself, with implantation posing a 15% risk of “minor and reversible problems”, and a 2–3% risk of fatal or hemorrhagic events, infection, lead fracture, and dislocation [28]. Between 2013 and 2017, there were 711 bilateral DBS placement surgeries in The Netherlands, a subset of which were suffering from PD. Of those 711 surgeries, 169 patients required the DBS system to be either replaced or removed entirely [29]. These side effects and adverse outcomes can partially be attributed to suboptimal placement of the DBS lead, which is dependent on the accuracy of the preoperative planning procedures [30,31].

## 2. Using MRI to Target the STN in PD for DBS

As noted, the success of DBS treatment is partly determined by the accuracy of targeting the STN. Further, targeting is dependent on stereotaxic precision, neuroimaging methods, and electrophysiological mappings [32]. Identification of the STN can be achieved in two ways: indirectly or directly. Indirect targeting refers to identification of the DBS target via application of reformatted anatomical atlases, formulae coordinates, and distances from anatomical landmarks. These standard targets can be applied to a patient’s individual magnetic resonance imaging (MRI), or can be used as a coordinate for navigation with a stereotaxic reference system (see next paragraph). Additionally, intra-operative microelectrode recordings, macrostimulation, and intraoperative behavioral feedback are commonly used for verification with indirect targeting [32,33]. Direct targeting refers to visualization of the STN on patient-specific MRI images [34,35].

For indirect targeting, the most common landmarks are the mid-way point between the anterior and posterior commissure (AC and PC, respectively), which are visualized and marked on a T1-weighted (T1w) MRI, computer tomography (CT), or ventriculography [33,36]. The native brain is commonly realigned to the AC-PC with a Euclidean transform [37,38]. This transform provides an augmented matrix with a 3D homogenous coordinate system, allowing for application of formulae coordinates and distances. The standardized STN coordinates are defined as 12 mm lateral, 4 mm posterior, and 5 mm inferior to the mid commissural point [39]. Some centers may utilize their own reference points, such as the top of the red nucleus [40,41,42].

Direct targeting with patient-specific MRI is generally preferred as the STN is known to shift with both age and disease, as well as vary in size, shape, and location across individuals [43,44,45,46,47]. Clinical MRI typically visualizes the STN using T2-weighted (T2w) images, which present the nucleus as a hypointense region relative to surrounding tissue. The optimal part of the STN is considered to be the ventral dorsolateral portion, also termed the somatosensory region, and is assumed to have direct connections with pre-motor cortical areas [48]. As with indirect targeting, direct targeting also incorporates AC-PC alignment, which provides the common reference system required for frame-based stereotaxic surgeries. Additionally, AC-PC alignment allows for comparisons between planned target location, actual target location, and postoperative verification. Therefore, clinical identification of the STN is usually achieved with a combination of both direct and indirect targeting methods.

The presence of extreme side effects and lack of clinical effect that can occur with DBS may arise from either direct or indirect targeting. One method for increasing the success and efficacy of DBS is to optimize preoperative planning procedures via neuroimaging techniques. For instance, advanced MRI can be used to increase visualization and understanding of anatomy, connectivity, and functioning of the STN. This information can then be used to inform on optimal electrode placement on a patient-specific basis.

The goal of this paper is to explain the current procedures for structural target identification of the STN for DBS in PD using MRI. We identify limitations that may contribute to suboptimal identification of the STN and provide alternatives for optimizing MRI in order to visualize the STN. The organization of topics is as follows: field strength; current procedures for intra and post-operative verification with microelectrode recordings; SAR limitations; shimming and magnetic field corrections; sequence types and contrasts; voxel sizes; motion correction; registration and image fusion; quantitative maps; complications unrelated to pre-operative planning; and conclusions. The suggestions are presented with the underlying expectation that more accurate visualization can translate into targeting and implantation with increased precision.

## 3. Field Strength

Pre-operative MRIs are obtained to both visualize the DBS target and to assess for potential comorbidity and identify venous architecture to ensure a safe entry route for surgery. The quality of MRI is dependent on a large number of factors. One of these factors is the signal-to-noise ratio (SNR), which is strongly influenced by field strength (Tesla or T for short) (see Figure 1) [24,49,50]. SNR can be defined as the difference in signal intensity, effectively determining the amount of signal that represents the true anatomy compared to noise and random variation [51,52]. Low-field MRIs such as 1.5 or 3 T are routinely used for DBS targeting. However, recently, an ultra-high field (UHF) 7 T MRI system has been approved for medical neuroimaging [53]. Compared to 7 T, 1.5 and 3 T MRI tend to suffer from both inherently lower SNR and low contrast-to-noise (CNR). CNR reflects the difference in SNR between different tissue types, which is therefore essential for specificity [54,55]. Moreover, the STN is an inherently difficult structure to visualize as it is a small structure located within a very deep and dense portion of the basal ganglia and is surrounded by structures containing similar chemical compositions. This is exemplified by vast inconsistencies in observed volumetric measures, size, and location estimates of subcortical nuclei reported at low field strengths [44,45,46].

The quality of the magnetic field is also determined by magnetic field gradients. MRI gradients are characterized by the change in the magnetic field as a function of distance. The MRI gradient arises from gradient coils, which are a set of electromagnetic components within the scanner that are used to control the magnetic field [56,57]. Weaker gradients arising from lower magnetic fields cause g-factor penalties, whereby an inhomogeneous B1 field causes artificial signal differences and noise amplification in tissues further from the coil in the subcortex at 3 T compared with 7 T MRI [58,59]. SNR is therefore lower in subcortical structures relative to the cortex due to the larger distance between the center of the brain and receiver coil elements. These differences are amplified at low field compared to UHF [60,61,62].

However, SNR scales supra-linearly with the static magnetic field, with up to a sixfold increase at 7 T compared to 3 T MRI [54,55]. This means that UHF-MRI can provide better quality images at a higher spatial resolution, increased contrast, and shorter acquisition times [51,63,64]. Reduced acquisition is essential, as clinical radiologists are often under strict time pressures that are intrinsically linked to value-based healthcare systems and cost-effectiveness rather than scientific value [65]. Numerous empirical studies and reviews have noted the advantages of utilizing UHF-MRI in clinical settings, performing direct comparisons between low- and high-field strengths for visualizing finer details of smaller nuclei, which are common targets for DBS [34,52,66,67,68,69,70].

Developments in array coil designs and parallel imaging techniques have resulted in the possibility to measure specific portions of tissue simultaneously. The simultaneous measurement increases SNR by a factor of 3 to 10 when compared to standard volume coils used at clinical field strengths, which are unable to selectively excite separate portions of tissue [60,63]. This is discussed in more detail later in the paper.

Importantly, there are caveats with regards to the implementation of UHF-MRI. Firstly, the Siemens 7 T MAGNETOM Terra is the only UHF-MR system to have obtained Food and Drug Administration (FDA) 510(k) clearance for clinical neuroradiology, and other applications of 7 T MRI are therefore considered experimental. Expense and accessibility is among the most important and most time-limiting factors in implementing UHF-MRI into clinical settings; less than one hundred 7 T systems exist worldwide, making up about 0.2% of all MRI systems [24,71]. Moreover, increased specific absorption rates (SAR), field inhomogeneities, local signal intensity variations, and signal dropout are factors that can reduce the benefits of 7 T MRI when not properly accounted for [72]. These can be countered with optimized shimming and pre-processing techniques such as bias field correction. However, these techniques require expertise that is not typically available within clinical settings [73,74,75].

## 4. Current Procedures for Intra- and Post-Operative Verification with Microelectrode Recordings

Current standard practices within The Netherlands includes both pre-operative planning with neuroimaging methods and intra-operative verification with microelectrode recordings (MER). In this case, once the target has been decided, the DBS system will be implanted in two steps. First, the surgeon will create a burr hole in the skull on both hemispheres. If microelectrode recordings (MER) are used, the MER leads will be inserted into predefined coordinates. In 0.5 to 2 mm intervals from around 10 mm above the target coordinate, MER will start recording activity through macrostimulation. Multiple MERs may be placed into the STN at around 2 mm apart within the anterior, posterior, central, medial, and lateral portions. The MER lead that outputs consistent oscillations of beta bursts that are indicative of STN activity will be selected for test stimulation and subsequent implantation. If the patient is awake, additional intraoperative behavioral testing may be performed to assess the therapeutic effect of specific stimulation programs. Once the target has been verified via intra-operative neuroimaging (CT or ultra-low field MRI), the leads will be permanently implanted and then connected to a cortical grid and a stimulator will be inserted under the chest [76,77,78,79].

Not all centers use pre-operative CT or MRI and instead rely on standard coordinates with MER verification (and vice versa). There are reports that suggest MER significantly improves DBS outcomes [80], and that MER fails to show any significant benefit compared to direct targeting [81]. Moreover, there remains a mismatch of around 20% in the planned target coordinate based on MRI, compared to the actual optimal location identified with MER when using 1.5 and 3 T [82,83]. Further, the use of intra-operative ultra-low field MRI for identification of the test leads during surgery has shown to be as effective as MER in improving post-operative motor symptoms [84]. Moreover, while not a strictly scientific issue, the application of MER more than doubles the cost of a bilateral STN surgery [85]. See [86] for an extensive overview on comparisons between MER and MRI for STN identification in PD.

Lastly, post-operative management requires the identification of optimal stimulation parameters. These parameters can vary per patient, and some patients may require DBS in combination with medication. Microlesioning effects and acute foreign body reactions can impact the homeostasis of STN function and lead to a misinterpretation of DBS efficacy. Therefore the patient should ideally be assessed several times at different stages after the surgery [87]. Baseline motor function is initially obtained after total withdrawal of dopaminergic medication [88]. Axial motor symptoms such as bradykinesia, rigidity, stability, gait, posture, and dysarthria are assessed with rating scales such as the Unified Parkinson’s Disease Rating Scale Part 3 (UPDRS, III) or Movement Disorders Society (MDS)-UPDRS [79,89]. As the DBS lead consists of multiple contact points, each point is tested separately through monopolar stimulation, beginning with a standard frequency of 130 Hz and pulse width of 60 µs [90]. Amplitudes are varied in a step-wise manner and the lowest amplitude that results in the highest suppression of clinical symptoms with the absence of sustained adverse effects will be chosen as the optimal stimulation parameters [27]. More in-depth literature on practices for post-operative verification, stimulation programming, and care can be found in [91,92,93] and the references therein.

## 5. SAR Limitations

SAR refers to the amount of energy deposited into the body due to the radio frequency (RF) pulses applied with MRI sequences. RF pulses are emitted via electrical currents through coils, being used to generate the B1 field [74]. RF deposition can result in tissue heating, and to ensure that the absorbed energy does not induce local thermal damage, there are SAR limitations based on the region of interest, with the amount of SAR depending on tissue type [94,95]. However, field inhomogeneities increase with field strength, as the RF wavelength scales according to the size of the object being imaged, which then reduces its ability to penetrate the brain with a uniform power [96,97]. In the case of UHF-MRI, stronger gradients are required to magnetize tissues in the middle of the brain and to create a homogenous field, which results in higher SAR. Therefore, the safety limits are reached sooner at UHF than with lower field. Moreover, SAR can vary person to person due to individual differences in anatomy. This means that scan acquisition can require real-time parameter adaptation. Maintaining a low SAR can be achieved by increasing the repetition time (TR), reducing the flip angle (FA), or by reducing the number of acquired slices. Unfortunately, introducing these parameter changes to MR sequences can negatively affect the quality of the scan [98,99]. This invites an ethical debate as to whether future FDA-approved sequences and image pre-processing methods at UHF would allow for such real-time deviations in a clinical protocol where SAR limitations are reached and sequence amendments are required.

Further, there are more absolute and relative contraindications at UHF including pacemakers, surgical implants and prosthesis, and foreign bodies, even if they are not metallic or comprised of diamagnetic materials due to potential local heating and subsequent torque and increased SAR. Moreover, in our experience, many DBS candidates may not be scanned due to site-specific criteria. For instance, while a non-metallic or non-paramagnetic dental bridge is not listed as a contraindication, the guidelines for the 7 T site at some locations required such patients to be excluded. Even more contraindications exist at 7 T, including circulatory and clotting disorders, which makes UHF-MRI less compatible with a larger portion of the elderly population, including the majority of PD DBS patients [100]. Therefore, optimizing 3 T remains a viable option where UHF-MRI cannot be applied. However, while 3 T may theoretically be optimized to allow for increased visualization of subcortical nuclei, it is essential to remember that acquisition times will be much longer than that of an analogous 7 T sequence [24,101,102,103]; this concept will be discussed throughout the paper.

## 6. Shimming and Magnetic Field Corrections

Shimming refers to the process of homogenizing either the main magnetic field (B0) or the radiofrequency field (B1). Inhomogeneity of the B0 field occurs when materials with different magnetic properties and susceptibility enter the bore, resulting in image distortion and signal loss. For example, the interface between brain tissue and air arising from the sinuses can cause artifacts within the frontal and temporal areas. These brain–air interface-induced artifacts can result in large shifts in the observed anatomical locations of nearby brain structures and cortical surfaces [104]. While post-processing techniques exist to correct some of these erroneous signals, they cannot control for complete signal loss and dropout. Therefore, the field needs to be shimmed prior to the acquisition of the main MRI scan.

Shimming the B0 field can occur passively by strategically placing ferromagnetic sheets within the bore itself to form the distribution of the magnetic field toward a more uniform state [105] or by using patient-related inserts such as an intra-oral pyrolytic carbon plate [106]. This process is useful for removing field imperfections related to hardware, although is not generally utilized in clinical practice as it is laborious, inflexible, and temperature-dependent. More commonly, the field can be actively shimmed, which uses currents within the MRI system to generate corrective magnetic fields in areas showing inhomogeneous signals [105].

Active shimming is limited by the ability to model and reproduce the distortions that occur within the field. Shimming is generally based on the principles of spherical harmonics (SH), which use orthonormal equations to index changes in signal waveforms representative of field inhomogeneity. The mapping and the correction of the inhomogeneity is achieved by superimposing the magnetic field with an opposing corrective field equal to and a reversal of the polarity within a spatial distribution deemed erroneous by the SH coefficients [107,108].

The order of SH is dependent on the number of dedicated current-driven coils. Traditional clinical and low-field MR systems will employ lower-order shimming methods mainly due to cost and space restraints [57]. Low-order shims primarily utilize linear terms including addition, scaling, and rotation of the SH coefficients to model the magnetic field. Linear SH coefficients function to resemble and compensate large-scale, shallow magnetic field components that can be corrected with a current offset applied with a standard gradient coil. This is typically achieved automatically with the use of a pre-scan B0 map. More local changes can be compensated for with dynamic shimming. However, this is most commonly used for multi-slice MR, which is prone to additional eddy current distortions and requires dedicated amplifier hardware. Further, the optimal shim method will depend on the desired contrast [109]. Ideally, each sequence should require an additional shim.

As field inhomogeneities increase with field strength, higher order harmonics are therefore required for UHF. Higher order SH allows for correcting more complex-shaped inhomogeneities by incorporating an additional non-linear quadratic field variation that allows for modelling the bending of curves in space. This requires supplementary dedicated shim coils, which can counter-intuitively induce additional distortions in the middle of the brain. Despite efforts to harmonize parameters, shimming is often site- and field-dependent, and manual iterative shimming is not always possible due to time constraints and/or limited expertise.

Additional B1 mapping is essential for accurate quantitative measures of signal intensities within the correct geometric space. Inhomogeneous B1 fields can result in distorted flip angles (FAs). FAs index the amount of net magnetization rotation experienced during the application of an RF pulse. If FAs are incorrectly calculated, geometric distortions occur, which reduces the accuracy in T1 and T2 values. B1 mapping allows for the correction of FA values prior to acquiring a structural scan. Primary B1+ mapping methods can be incorporated into sequence acquisition. This is most commonly achieved with the double angle method (DAM), which estimates local FAs from the ratio of two images obtained with different FA values. An additional 3D multi-shot method can be incorporated, which uses non-selective excitation to minimize inhomogeneous spin excitation across slices. Alternatively, spoiled gradient echo (GRE) sequences with variable FAs (VFA) and actual FA imaging (AFI) are commonly employed, which sample multiple T1 values to simulate signal differences across tissues [110,111,112,113].

Pre-processing of gradient non linearities (GNL) and intensity non-uniformity with retrospective image-based interpolation is also possible. Corrections for GNL are rarely accomplished in clinical settings but are commonplace for research-based applications. The magnitude of GNL increases with distance from the isocenter and can cause the visualization of structures to shift by up to 5 mm, which is detrimental for preoperative planning [114]. Correcting for GNL can be achieved by incorporating a low-pass filter to remove smooth spatially varying functions. Other GNL correction schemes include surface fitting and feature matching that rely on intensity-based methods. Intensity-based methods assume that different tissue intensities do not vary significantly unless they are subject to an erroneous bias field, where variations within one area can be corrected from the field of another spatial location within the image. Alternatively, histogram-based methods use a priori knowledge and manual input of known intensity and gradient probability distributions to correct images. B1 corrections can be achieved offline via image pre-processing steps with the FMRIB Software Library (FSL), Statistical Parametric Mapping (SPM), or Advanced Normalization Tools (ANTs) [115,116,117,118]. However, these methods must be considered experimental and their use in image correction for MRI in pre-operative planning is not currently FDA-approved.

## 7. Sequence Types and Contrasts

### 7.1. T1

As discussed, accurate DBS implantation requires careful trajectory planning and identification of vasculature to limit the risk of hemorrhagic complications. Visualization of larger venous architecture is most commonly achieved with an anatomical T1w scan with added gadolinium [119,120]. In its most basic form, T1w can be viewed as an anatomical scan that approximates the appearance of macroscopic tissues. T1w will visualize white matter as hyperintense; fluid, e.g., cerebral spinal fluid (CSF) as hypointense; and grey matter at intermediate intensity. A T1w contrast is achieved with a short echo time (TE) and repetition time (TR) and is a function of the longitudinal relaxation time, referring to the time it takes excited protons to return to their equilibrium subsequent to the application of an RF pulse. T1 is more sensitive to fat and fluid and therefore provides excellent differentiation between grey and white matter. Additional intravenous contrast agents will cause the recovery of the longitudinal magnetization of blood to quicken and therefore increase further contrast between veins and white matter [121,122,123]. For visualization of venous architecture, some centers may use any or a combination of T1w structural imaging, or they may use post-processing techniques such as susceptibility weighted imaging (SWI) and venography, which can be created from GRE-based sequences with flow compensation, or time-of-flight angiography. These types of sequences apply multiple RF pulses with short TRs to over-saturate static tissues and therefore suppress their signal, causing moving components such as blood to appear more hyperintense [124,125,126]. T1w MRI can also be used to rule out co-morbidities such as oedema, tumors, or other brain pathologies. See Figure 2 for an example of different contrasts.

### 7.2. T2

T2w images visualize grey matter as intermediate intensity and white matter as hypointense, although deep grey matter structures can appear even darker depending on the ferromagnetism of their tissue composition. As mentioned, visualization of STN is traditionally achieved with T2w sequences [127,128,129]. T2w MRI represents transverse relaxation, referring to the amount of time it takes excited protons to lose phase coherence. This dephasing is a tissue-specific process and takes longer for areas with high paramagnetic metal deposition such as iron. As the STN is iron-rich, the contrast is increased, and the nucleus appears hypointense compared to white matter tracts and surrounding grey matter structures. Typically, T2w contrasts within the clinic will come from fast-spin echo sequences that have both a long TE and TR, which are relatively immune to magnetic susceptibility artifacts. However, there is no general consensus as to the optimal sequence required for prime STN imaging. Theoretically, various sequences can achieve the same weighting but vary significantly in terms of their ability to accurately visualize the STN [130]. Moreover, the type of sequence will depend on the field strength, and contrasts are not always analogous across, for instance, 3 and 7 T [131]. Similarly, different MRI vendors will supply similar contrasts via sequences and sequence parameters with different names, making it difficult to draw comparisons between them [50,132,133].

### 7.3. T2* and Susceptibility-Based Contrasts

Traditional clinical T2w sequences suffer from low signal and contrast. An alternative contrast that can be used to image the STN directly comes from 3D gradient echo (GRE) sequences, which can be used to create T2* images. Typically, GRE sequences will include a low FA, long TEs, and long TRs. Moreover, gradients are applied to initiate dephasing, as opposed to an RF pulse in traditional spin echo sequences [109,134]. These gradients do not refocus field inhomogeneities such as RF pulses do. Therefore the T2* contrast arising from GRE reflects magnetic field inhomogeneities caused by the dephasing of neighboring areas that occurs at different rates, and further interact with the signal of adjacent voxels [135]. As GRE sequences assess macroscopic intervoxel and microscopic intravoxel magnetic susceptibilities, it is important to adapt sequence parameters according to the tissue of interest [136]. The tissue characteristics of the STN undergo PD-specific changes, such as dopaminergic denervation and excessive iron deposit, which require adjusted parameters such as TE for optimal contrast [137,138]. Similarly, iron increases with normal aging requires different adaptations to TEs [139]. GRE sequences also incorporate multiple echoes to account for differences in magnetic susceptibility across tissues. Further, susceptibility effects are stronger for smaller voxel sizes as the dephasing is reduced [135]. This makes T2* imaging more appropriate for higher field strength MR, as smaller voxel sizes can be achieved with faster acquisition times [130,140]. These T2* images can be further processed to create quantitative maps that will be discussed in later sections.

Alternatively, susceptibility weighted images (SWI) can be created from T2*-based sequences by independently processing magnitude and phase images. Magnitude images reflect the overall MR signal, and their corresponding phase image contains information about field inhomogeneity, differences in local precession frequencies, and motion [141]. Phase images were largely discarded before the implementation of SWI as they require complex unwrapping, referring to the extraction of their original numerical range, which is constrained in the outputted image to [−π, +π] [142]. However, phase can be used to visualize information that would otherwise be barely visible in magnitude images. Small structures result in field variations with high spatial frequencies, which can be used to enhance contrast by applying a high pass filter. The resulting SWI image is the product of multiplying the phase mask with the magnitude image [142,143,144]. It remains somewhat controversial to what extent SWI signal increases from 1.5 T to 3 T MRI. Moreover there is little evidence for increased accuracy for SWI at 3 T compared to classic T2 imaging [145]. However, SWI is significantly more accurate compared to traditional contrasts at higher field strengths [146,147,148]. GRE-based sequences and T2* contrasts can provide more detail regarding the shape, surface, and location of the STN compared to standard T2w spin echo-based sequences. This could translate to more accurate DBS targeting if it were used for preoperative planning. Improvements can refer to a smaller deviation between planned and actual lead location, a reduction in reimplantation or removal requirements, increased clinical efficacy, or decrease in associated side effects. However, the use of T2* contrasts and UHF-MRI remains widely debated and requires further validation [37,70,144,148,149,150].

We attempted to use a T2*-based UHF-MRI with a GRE-ASPIRE sequence [151] on a 7 T Siemens MAGNETOM system (Siemens Healthcare, Erlangen, Germany) for STN DBS planning in PD patients. The 7 T T2* scan consisted of a partial volume covering the subcortex, obtained with multiple echoes (TE1–4 = 2.47, 6.75, 13.50, 20.75) and 0.5 mm isotropic voxel sizes in just under 8 min. This was overlaid with a 3 T T2w turbo field echo sequence obtained on a 3 Tesla Phillips Ingenia system, with a single TE of 80 ms and voxel sizes of 0.45 × 0.45 × 2 mm, and an acquisition time of around 6 min. When merging the 3 T and 7 T data, the STN appeared elongated along the posterior direction on 7 T. The optimal target coordinate appeared more superior, posterior, and lateral on the 7 T image than the optimal coordinate on 3 T. Here, the 7 T coordinate was used as the posterior test site sampled with MER was used as a target for DBS surgery. Typical STN activity was not observed, although intraoperative behavioral testing revealed that patients would exhibit a beneficial clinical effect. Such a finding may be explained by the fact that the test electrode was instead stimulating white matter fibers exciting the STN, such as the fasciculus lenticularis or medial fiber bundles. It is, however, unclear as to whether this discrepancy in optimal STN coordinate is due to errors in registration across field strength, smoothing factors and interpolation automatically applied by the pre-operative planning system that reduced the resolution of the 7 T data, magnetic field inhomogeneity, or geometric distortions of the T2* image. The issues regarding image correction and manipulation are discussed in later sections. It is entirely plausible that the discrepancy in optimal target location across field strength was due to human error, and the operating surgeons perhaps were not used to interpreting the high-resolution susceptibility-based images. Therefore, factors other than contrast and sequence type can influence the usability and accuracy of susceptibility-based imaging for neurosurgical applications.

It is important to note that the sequences described in this specific instance are not standardized across centers, and scanner vendors, field strengths, contrasts, and sequence parameters, even within the same sequence type, will differ across DBS centers and research institutes. This makes a direct comparison across the quality and replicability of MRI scans very difficult, and unless systems are harmonized, interpretations should be site-specific. See [86,130] for a comprehensive review on sequences used for imaging the STN.

### 7.4. Multi-Contrast MRI

Multi-contrast sequences may offer a novel alternative for eliminating the requirement of registration and resampling of separate scans while simultaneously reducing scan acquisition time (Figure 2) [152]. A recently developed multiparametric imaging sequence is the Multi Echo (ME) MP2RAGE, which is largely unaffected by B1 inhomogeneities [153,154,155,156,157,158,159,160]. This allows for the acquisition of T2*-based contrasts from which subsequent SWI and quantitative susceptibility maps (QSM) can be created in the same space as the T1 images [158,160]. Other benefits of multiple contrasts is that they contain complimentary information that can be used to jointly denoise and improve the SNR of the acquired images [161,162,163].

## 8. Voxel Sizes

Clinical T2w images often incorporate anisotropic voxel sizes with large slice thickness in the z direction. This allows for higher in-plane resolution along the axial plane, which is primarily used for targeting (Figure 3) [145,149]. Voxel sizes will typically range between 0.45 × 0.45 × 2 mm and 1 × 1 × 3 mm. Lower resolution allows for shorter acquisition times of around 5 min, simultaneously limiting the effect of artifacts due to subject movement. However, anisotropic voxels suffer from partial voluming effects (PVE), which refer to the blurring of signals across voxels, resulting in averaging different tissue types and reducing specificity [165]. PVE are especially problematic for small structures such as the STN. Volume estimates are commonly used as an index of scan quality, and have shown consistent deviations of more than 50% from ground truths when slice thicknesses were three times the size of the alternate planes [166]. Moreover, anisotropic voxels will decrease the accuracy of resampling to super resolutions, which is an automatically incorporated step of pre-operative planning systems [167].

As spatial resolution is dependent on voxel size; smaller voxels should allow for more detailed and finer grained visualization of smaller structures. Voxel sizes can be reduced by increasing the acquisition matrix, reducing slice thickness, or decreasing the field of view. However, these factors can each negatively affect the SNR. The loss of SNR can be compensated by simply including more repetitions per sequence, which is an issue for PD populations as it necessitates an increase in acquisition time and requires the patient to be still. However this is often not possible for patients with movement disorders [166]. The loss of SNR caused by decreasing voxel sizes at lower fields can be counteracted through the use of UHF-MRI [130].

When targets in clinical MRI are verified with MER, the large slice thickness means that the spatial resolution is penalized along the *z*-axis. Therefore the depth of the electrode cannot be optimally planned and electrophysiological samplings are conducted to identify the ideal electrode placement [32,38,40]. This testing often requires that the patient is awake and endures behavioral assessments, which are stressful and physically demanding, prolong the time of the surgery, and can increase the risk of infection or hemorrhaging [168,169,170]. If smaller voxels can increase spatial resolution, three-dimensional anatomical accuracy, and tissue specificity, the requirement for intraoperative microelectrode recordings, multiple test electrode implantations, and awake behavioral testing could be eliminated, ultimately increasing patient comfort and reducing operation time.

However, voxels with a sub millimeter isotropic resolution used purely for identification of DBS targets, rather than for instance venous architecture, may not directly improve targeting accuracy. This is because the spatial resolution of stereotaxic coordinate systems is around 1.2 mm and chronically implanted conventional DBS electrodes are larger than 1 mm [171]. In addition, segmented DBS leads with directional steering may offer increased spatial resolution when recording local field potentials compared to traditional omnidirectional contacts [172,173]. Further, the development of microscale DBS contacts via multiresolution electrodes would allow for finer control of the stimulation volume and more precise targeting of smaller regions, matching the order and spatial resolution of submillimeter resolution MRI [174].

## 9. Motion Correction

Generally, clinical imaging for preoperative planning for DBS does not correct for motion, and the scans do not tend to incorporate acceleration methods such as parallel imaging. Accurate imaging requires the subject to remain still. If a patient scan exhibits severe motion artifacts, the scan is simply run again. MR images can be distorted by multiple sources of motion arising from breathing, cardiac movement, blood flow, pulsation of cerebrospinal fluid, and patient movement [104]. This can cause distortions in the image such as ghosting, signal loss, and blurring, as well as Gibb’s and chemical shift artifacts [175]. Such artifacts can mask or simulate pathological effects [104]. Motion artifacts are particularly prevalent when imaging patients with movement disorders but can be controlled for in a number of ways such as timing medication to be most optimal during the time of scanning or administering additional sedatives during the scan. Moreover, the head and neck should be supported with pads to improve patient comfort, which will also limit movement.

The most logical method of limiting motion artifacts is to decrease the acquisition time. Sequence paraments can be manipulated to shorten the acquisition time by obtaining larger voxel sizes, a partial field of view (FOV), incorporating simultaneous multi slice 3D imaging and parallel imaging techniques, signal averaging, or obtaining multi contrast images. To correctly utilize these potential solutions, each factor should be considered relative to one another. For instance, partial FOVs can induce aliasing, fold over artifacts, and reduce the SNR, which can, to a certain extent, be countered by isolating the excitation to a localized region by using either multiple pulses, signal averaging, or fat suppression methods. Contrary to this, it may increase the effects of field inhomogeneity, but be combated with factors such as spatial pre-saturation. Such issues highlight the dynamic nature and interplay of sequence parameters and hardware, which can be largely overcome through the use of stronger field strengths such as 7 T.

Parallel imaging (PI) is a reconstruction technique rather than a sequence commonly employed to accelerate acquisition time [176]. Magnetic resonance (MR) images are not directly collected but are instead stored in a Cartesian grid, representing a spatial frequency domain known as k-space. K-space data is collected via superimposing spatially varying magnetic field gradients onto the main magnetic field [55,177]. Generalized auto-calibrating partially parallel acquisition (GRAPPA) methods speed up acquisition n time under-sampling each line of k-space in the phase-encoding direction. Additionally, partial FOVs are collected independently, corrected, and then reconstructed within the frequency domain [178,179,180]. Alternatively, sensitive encoding methods (SENSE or ASSET) can shorten scan times, and these methods occur in the image domain where data are obtained using multiple independent receiver channels where each coil is sensitive to a specific volume of tissue, which is then unfolded and recombined to form the MR image [177]. However, PI methods are associated with a number of artifacts including ghosting, speckling, wrap around, and g factor penalties and ought to be used with caution [181,182,183].

Motion correction can be conducted prospectively in real time by updating the image geometry during the scan, or retrospectively by post-acquisition registration techniques and manipulations during image reconstructions [184]. Additional hardware is required for prospective methods that are implemented within the scanner itself. In this case, fiducials can be attached to the patient’s head, which assesses the extent of movement and adjusts the gradients accordingly. Alternatively, you can employ optical tracking or reflective markers, which are linked to a camera inside the bore. Motion correction is then achieved by either re-registration slice-by-slice during the scan, adjusting first order shims, and/or varying the gradient system online [185,186]. As discussed, motion artifacts do not have to come from patient movement but can arise on a much smaller scale at the proton level. Protons in blood, for example, exhibit a non-static magnetic field due to the variation of gradients in space. That is, they can miss rephasing pulses and therefore decay in signal before it can be read out by the scanner, especially for spin echo sequences that are used for obtaining T2w images [187]. This phenomenon is known as flow-related dephasing and results in artifactual phase shifts and signal distortion. In some instances, this can be useful, for example in angiography sequences, the negative effect is larger in sequences with longer TEs, such as those required for accurately imaging the STN. Adding in flow compensation or gradient moment nulling, which applies additional gradient pulses prior to the signal readout to compensate for signal decay, can compensate for this dephasing [188,189]. However, this is a computationally heavy process and is largely only suitable for partial FOVs. Alternatively, the sequence may be synchronized so that the acquisition occurs in time with the cardiac or respiratory cycle, which is known as cardiac gating and simultaneously requires pulse recordings or electrocardiograms [104].

## 10. Registration and Image Fusion

Using MRI to visualize deep brain structures such as the STN for DBS is a multi-stage process that involves the acquisition of multiple separate contrasts that require registration to a common, patient-specific native space. For pre-operative planning, at least two sets of image registrations are required: (i) anatomical T2 to T1 and (ii) pre-registered anatomical T1 and T2 to stereotaxic space defined by the CT or MRI including the coordinate frame. In this section, we focus on registration and fusion of MRI. For literature including alternative imaging modalities such as CT and ventriculography, see [190,191].

Image registration refers to the process of aligning a moving source image onto a fixed target through an estimated mapping between the pair of images. While the exact parameters incorporated within pre-operative planning systems are mostly proprietary, the general process will require a rigid registration, defined by six parameters: translation and rotations along the *x*-, *y*-, and *z*-axes. This refers to the spatial transformation of how a voxel can move from one space to another [192]. Transformations also require additional parameters such as interpolation and cost function. Interpolation refers to the process of re-gridding voxels from the source image to the target, an essential procedure as each pixel within the transformed image may not represent a whole integer within the target image. This is especially true when T2w images consist of anisotropic voxel sizes and the T1 images are isotropic. Therefore, the goal of interpolation is to piece back together the voxels that have been moved. Clinical neuroimaging traditionally employs the simplest intensity-based methods such as nearest neighbor interpolation, also known as point sampling, which assumes that similar values in different images are closer together and therefore constitute the same location [193,194]. Cost functions are used to assess the suitability of a given transform. This can be achieved with either similarity metrics such as mutual information, which compares, on the basis of pixel intensities, the differences between the transformed source and target image [195]. These registration steps are all conducted automatically within pre-operative planning systems, with the only manual alterations relating to viewing criteria such as brightness and intensity. This is suboptimal, as registrations often need tweaking and optimizing, and it becomes challenging to suggest exact methods for optimizing registrations with regards to pre-operative planning systems as it remains unclear as to what exact parameters are employed.

Linear within-subject registrations typically employ intensity-based similarity metrics, matching images on the basis of intensities or intensity distributions. Intensity methods can be optimized to incorporate local patches that account for textures and geometric information that are missed when assessing for global identical intensities. An example is boundary-based registration, which forms the basis of intra-subject registration of T2 to T1 images within the Human Connectome Project minimal processing pipeline [196,197]. Registrations could be optimized to include an additional affine transform that incorporates scaling or sheering [198]. Alternatively, deformable registrations via attribute matching and mutual saliency (DRAMMS) can be achieved. DRAMMS applies confidence weightings for matching voxels across contrast and will relax deformation in local regions where contrast-specific tissues are mutually exclusive to image type. DRAMMS has proven useful in accounting for pathology, subcortical structures, and cortical thinning, which are all factors to consider when imaging PD patients [199].

Further, no quality or standardized evaluation for registration accuracy currently exists in clinical neuroimaging beyond subjective visual assessment. This is problematic as it becomes unclear as to whether the initial rigid body transforms are an accurate spatial representation of individual anatomy, which, if erroneous, could result in targeting errors and DBS lead placement. The gold standard of accuracy is instead dependent on the stereotaxic frame, which is an extrinsic marker and does not include information directly related to the MR image.

Medical imaging often incorporates automated image fusion, which refers to the process of aligning, resampling, smoothing, and combining the information of multiple images into a more informative and descriptive output; for instance, by combining T1 and T2 into a single image. Fusion occurs after registration with the goal of interpolating and smoothing MRI images to make them more visually appealing, which can theoretically recover a signal within the data despite the noise [200]. However, smoothing and resampling voxel sizes will reduce anatomical variability and location accuracy as they can include signal from neighboring structures, leading to an erroneous increase in the size of the nucleus and PVEs [166,201]. Such smoothing methods may not be compatible with quantitative images such as T2* maps and QSM, as these images represent distinct signal intensities of specific voxels that are outside the predefined values of the planning system. In effect, this could be a simple viewing error, rather than a total incompatibility.

## 11. Quantitative Maps

Broadly speaking, MR contrasts are driven by how much T1 or T2 signal contributes to the image. These T1w or T2w images are qualitative in nature and fail to accurately assess tissue parameters such as recovery or relaxation time. However, certain sequences allow for parametric mapping (quantitative MRI or qMRI), where the intensities within each pixel are proportional to the T1 or T2. These values can be used to quantify intrinsic, biologically meaningful tissue information [202]. Additionally, qMRI allows for direct comparison across time, across subjects, and across scanners or sites, which is essential for the development of neuroscientific research and its application to the clinical situation [203]. Moreover, quantitative measures can aid identification and visualization of target structures with an objective approach and can minimize human error resulting from subjective interpretation. qMRI can only be made from specific sequences that comply with the principles of differential weightings, which incorporate an inversion or saturation recovery parameter with multiple inversion times or spoiled gradient echo sequences with variable flip angles [204]. However, in our experience, quantitative sequences at 3 T take at least twice as long as weighted MRI sequences used in clinical settings.

As mentioned, quantitative maps are used to index anatomical composition. For instance, the observed relaxation of T1 is extremely fast in myelinated white matter. The inverse of longitudinal relaxation rates, known as R1 [205], is thought to be linearly related to myelin concentrations [206,207]. T1 maps have been utilized clinically, for example, with quantifying perfusion; imaging hemorrhages and infarctions; evaluating contrast uptake; monitoring of tumors, gliosis, and multiple sclerosis lesions [205,208,209]. Quantitative T1 maps will usually require post-processing, most commonly achieved with the look-up table method, which functions to relate pixelwise T1 values within the native map with predefined and validated intensity values [159]. The automated creation of these T1 parametric maps can be built into the sequence at a cost of both time and capacity. Further post-processing is often required and relies on expertise that is again typically not available within a standard clinical setting [160,210].

For DBS of the STN, T2* maps can be used to improve visualization of the STN because iron content causes the T2* relaxation time to shorten, which for the STN at 7 T is around 15 ms [61,211]. A frequently used method to create T2* maps is done by fitting an exponential decay curve to the signal intensities per pixel from each of the multiple echoes obtained from a GRE sequence [212]. Moreover, the pixel intensities of reciprocal T2* maps (R2*) are proportional to iron load, with STN R2* values hovering around 67 s^−1^ (1/15 ms) at 7 T [155,213,214,215,216]. Alternatively, T2* images can be post-processed to create quantitative susceptibility maps (QSM), which quantify a tissue’s magnetic susceptibility distribution on the basis of its perturbation of the magnetic field [213]. They are similar to SWI in that they are made from the separate magnitude and phase images of a GRE sequence, but they comprise multiple echoes and allow for quantitative measures rather than weightings. QSM requires initial phase unwrapping, background field extraction, and calculation of locally generated phase offsets, which refer to the fact that the phase of a single voxel can be expressed as either positive of negative, depending on its orientation relative to the magnetic field [214]. These phase-offsets are then deconvolved, typically with a dipole kernel, from which the underlying tissue susceptibility can be extracted per voxel, independently of surrounding voxels [215]. Moreover, QSMs are preferred over SWI, as SWI is limited by the non-local orientation-dependent effects of phase, which means that the same tissues can appear with different intensities on the basis of their location, whereas QSM solves this problem by convolving dipole fields [216,217]. Background removal methods based on principles of sophisticated harmonic artifact reduction for phase data (SHARP, also known as spherical mean value (SMV) filtering) and projection onto dipole fields (PDF) are commonly employed. SHARP is based on a theory similar to shimming, in that static magnetic fields and the corresponding phase maps are represented by harmonic functions. In regions of inhomogeneous susceptibility, the field will be non-harmonic, and background fields that are harmonic are eliminated from the phase data by subtraction [213,218]. The PDF method removes background fields by comparing the magnetic fields of dipoles inside a region of interest with those directly outside [219,220]. Alternatively, Laplacian boundary values can be used, which are based on a finite difference scheme [221]. However, quantifying an arbitrary distribution of susceptibility from the phase signal is challenging and poses an inverse problem whereby effects are first calculated from which parameters or causes are then determined, resulting in a noise amplification of the ensuing signal. The inversion problem can be solved with calculation of susceptibility through multiple orientation sampling (COSMOS). However, this method requires the acquisition of multiple head orientations, which is time-consuming and impractical for clinical use [222,223]. Morphology-enabled dipole inversion, or MEDI, will match the boundaries of each dipole with those observed in the T2*-weighted magnitude images [222]. Quantitative susceptibility and residual mapping (QUASAR) accounts for biophysical frequency contributions, which acknowledges that the notion that the local Larmor frequency is affected by the static field perturbations related to tissue susceptibility, as well as the magnetic field, chemical shifts, directional alignment of axons, and energy exchange between water and macromolecules [224]. Alternatively, some algorithms solve the entire equation within a single step by incorporating SHARP principles with simultaneous total generalized variation (TGV)-regularized dipole inversion [225,226]. Similarly phase removal using the Laplacian operator (HARPERELLA) simultaneously combines phase unwrapping and background removal [227]. These methods comprise tool boxes that are largely available in Matlab or Python (see [228] and the references therein).

The clinical potential of QSM lies in its sensitivity to variations in iron stored in ferritin and hemosiderin, lipids and calcium, levels of differential oxygenation-saturation present in venous blood, and identification of sub millimeter white matter microstructure [229,230,231]. Further, QSM has been shown to be superior to T2* in parcellations of the STN, which could translate into better visualization and targeting for DBS [223,228,232]. T2 relaxometry has been shown to predict motor outcome in some PD patients with STN DBS, where patients who have low T2 values may fail to show a clinical benefit [233]. It is possible that this can be explained by the fact that patients with low T2 relaxometry will have less contrast between the STN and the surrounding tissue, hindering the accurate visualization and targeting of the structure, which could be solved by employing QSM. However, QSM obtained during a scanning session is still experimental and under development. Further, there are many competing post-processing methods for creating QSM images, which makes translation challenging.

## 12. Complications Unrelated to Pre-Operative Planning

Lastly, we would like to mention that while this paper specifically refers to suboptimal placement of DBS leads due to the limitations of neuroimaging, negative outcomes of DBS application can arise independently of planning procedures and surgical expertise. For example, neurosurgery has been linked to brain deformation and shift, changes in cerebral spinal fluid volume, and intracranial pressure, which may induce spatial variability both during the surgery and cause a shift in the implanted lead location during recovery [27,234]. Similarly, DBS surgeries are associated with infection (mostly found in the chest and connector) [235]; reactive gliosis and gliotic scarring [236]; hemorrhage either during the surgery or delayed (in less than 5%) [237]; and, although rare, cerebral pneumocephalus [238]. In all these cases, the DBS system may require reimplantation, replacement, or removal.

## 13. Conclusions

In this paper, we have discussed some of the differences in current clinical MRI practices with optimized and UHF-MRI methods commonly employed in research environments. Clinical MRI hinges on weighted imaging with anisotropic voxel sizes and maintaining short acquisition, therefore being limited in signal and resolution. These current clinical practices are FDA-approved and are therefore deemed acceptable for neurosurgical purposes. However, the presence of side effects and non-responding patients nonetheless exist. Optimized 3 T and UHF-MRI tend to incorporate isotropic high-resolution imaging with quantitative and susceptibility-based contrasts for better visualization of deep brain structures, which, however, require more complex pre-processing and longer scan durations. The limitations incurred regarding reduced signal in clinical MRI and increased acquisition time with optimized 3 T can be largely overcome with the use of UHF-MRI. However, many of the image registration, correction, and post-processing techniques will typically require expertise that is outside the realm of traditional clinical settings. Importantly, the use of UHF-MRI and alternative contrasts such as QSM can only be implemented once pre-operative planning systems allow for their compatibility, which will require further FDA approval, not only for the MRI system but also for specific sequences. Additional approval for clinical use may be required for pre- and post-processing, such as the algorithms used for registration or calculation of quantitative maps.

We therefore propose that where UHF-MRI is not accessible, higher quality imaging can be obtained with optimized 3 T, although this will take longer than is perhaps clinically feasible for patients with severe movement disorders. Continued direct collaboration and combined efforts between fundamental neuroscience researchers and clinicians will be essential for the development of optimized 3 T and UHF-MRI in the pre-operative planning process for DBS of the STN in PD. Multi-site clinical trials can facilitate the optimization and validation of certain sequences. Sequences with identical parameters should be compared on identical MRI systems and different sites to ensure harmonization and reliability, as well as to validate the desired sequences. Rates of deviations between planned and actual target locations should be compared across vendors and systems as well as across sequences. Similarly, access agreements to work-in-progress protocols from MR vendors would facilitate the development and optimization of sequences, and would open access to underling algorithms and adjustable parameters within to pre-operative planning software vendors (e.g., Medtronic, St. Judes, Brainlab, Abott, Nextim, and Boston Scientific).

Of note, while this paper focused specifically on the STN as the most popular target for DBS in PD, alternative targets also exist (for example, see Figure 1). Some centers have long preferred the internal segment of the globus pallidus, and more recent research is being conducted on the suitability of alternate areas such as the ventral intermediate nucleus or the pedunculopontine nucleus for DBS targets. For a more in-depth review, please see [10,239,240] and the references therein.

## Figures and Tables

**Figure 1 jcm-09-03124-f001:**
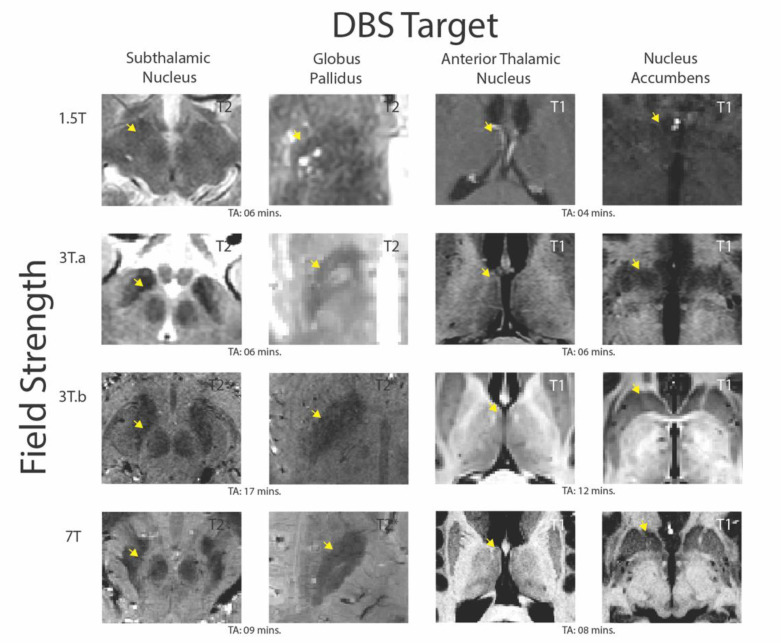
Visualizing deep brain stimulation (DBS) targets with different magnetic resonance imaging (MRI) field strengths (adapted from [24] illustrating DBS targets across field strengths, requiring different contrasts. We obtained 1.5 T images from a 52-year-old male Parkinson’s disease patient at the Maastricht University Medical Center (MUMC). Clinical 3 T and 7 T images were obtained from a from 57-year-old male Parkinson’s disease patient at the Maastricht University Medical Center (MUMC), and the optimized 3 T images were obtained from a healthy male age-matched subject at the Spinoza Center for Neuroimaging, Amsterdam. All images are shown in the axial plane and are present in their native space with no post-processing to replicate the visualization of each nucleus as performed on neurosurgical planning software. The T1 contrasts show the anterior thalamic nucleus and nucleus accumbens at all field strengths. The subthalamic nucleus and globus pallidus (GP) are shown with a T2 contrast at 1.5 T and clinical 3 T scan. Note that in the 7 T contrast, the medial medullary lamina is visible, allowing us to distinguish between the internal and external segment of the GP. For optimized 3 T and 7 T, the STN and GP are shown with a T2* contrast. The acquisition times (TA) for each scan are included to highlight the fact that optimized 3 T can provide high-quality images similar to those at 7 T but take nearly twice as long to obtain. While the STN and GP are visible in both 3 T images, the contrast and sharpness of borders increases at 7 T.

**Figure 2 jcm-09-03124-f002:**
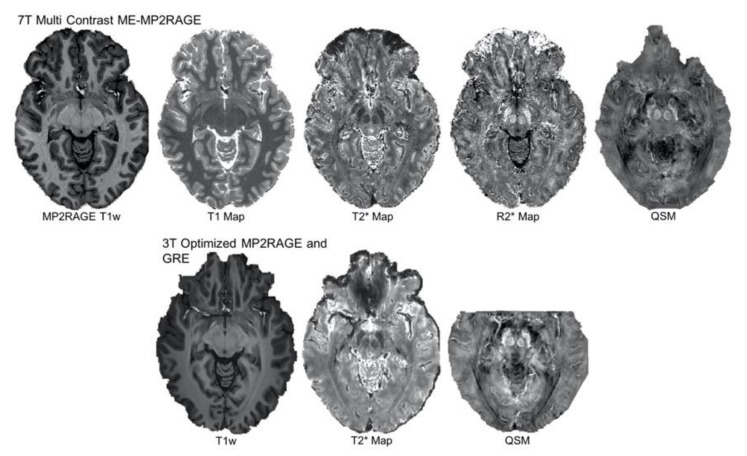
Multi-contrast imaging. The top row shows MP2RAGE T1-weighted, T1, T2*, and R2* maps and a quantitative susceptibility map (QSM) image obtained at 7 T within a single multi echo (ME) MP2RAGE sequence. Below are a 3 T T1-weighted map, a T2* map, and a QSM image, where each T1 and T2* were obtained with different sequences but were optimized to provide a contrast comparable to those obtainable at 7 T but without the inversions required for creating T1 maps. Both the 3 and 7 T images came from the same subject and are shown in the axial plane. The contrast and visibility of subcortical structures is indeed comparable across field strengths [164].

**Figure 3 jcm-09-03124-f003:**
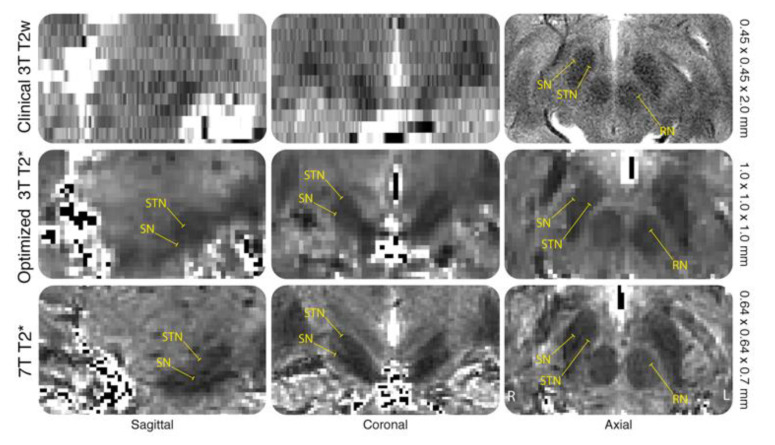
The effects of voxel geometry on the visualization of subcortical structures (adapted from [164]). Figure 3 shows clinical 3 T T2-weighted (T2w) with 0.45 × 0.45 × 2.0 mm voxel sizes, optimized 3 T T2* with 1.0 × 1.0 × 1.0 mm voxel sizes, and 7 T T2* maps with 0.64 × 0.64 × 0.7 mm voxel sizes. All images are acquired from a single subject and are shown at approximately the same anatomical level. The subthalamic nucleus (STN) and substantia nigra (SN) are shown at sagittal, coronal, and axial planes, with the red nucleus (RN) also highlighted in the axial plane. The anisotropic nature of the sagittal and coronal planes on the clinical 3 T do not allow for identification of any structure.

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
