# Peer review of "Methodological Considerations for Neuroimaging in Deep Brain Stimulation of the Subthalamic Nucleus in Parkinson’s Disease Patients"

_jcm, 2020, doi:10.3390/jcm9103124_

Round 1

Reviewer 1 Report

Isaacs et al re-submit an amended manuscript with some minor changes which are noted.

This final draft is certainly publishable in my opinion and i have no further comentary to add. I congratulate the authours on their work.

Author Response

Dear Reviewer 1, 

We are pleased to hear that you find our last revision acceptable for publication. Thank you for your constructive comments and helping us to create a more coherent paper. 

On behalf of my co-authors, again many thanks and all the best,

Bethany

Reviewer 2 Report

Appropriate modification have been performed in the manuscript. I have no further comments.

Author Response

Dear Reviewer 2, 

We are pleased to hear that you find our last revision acceptable for publication. Thank you for your constructive comments and helping us to create a more coherent paper. 

On behalf of my co-authors, again many thanks and all the best,

Bethany

Reviewer 3 Report

Dear Authors,

I see that you carefully considered recommended revisions after the initial review. In particular, your clarification about the organization of the paper was very helpful and made it easier to follow on the second read. In reading the revised version of the paper, there were still several areas which I feel can benefit from additional editing/revision:

1. Abstract: Consider re-wording “These negative responses to treatment can partly be attributed to suboptimal pre-operative planning procedures via direct targeting through Magnetic Resonance Imaging (MRI). One solution for increasing the success and efficacy of DBS is to optimize preoperative planning procedures via sophisticated neuroimaging techniques such as  MRI…” It is somewhat confusing that first sentence attributes suboptimal planning to MRI, and then the next sentence proposes MRI as the solution. Perhaps clarify that optimization of parameters, sequences, analyses etc. of the MRI is the solution rather than the MRI itself.

2. Introduction, paragraph 2 still needs significant rewording to clearly explain motor fluctuations and dyskinesias in Parkinson’s disease, and their relationship to medication – or, consider significantly simplifying/editing this section. Furthermore, a few sentences are not accurate or are misleading. For example:  “At this stage, the reliability, consistency and durability of the response to levodopa or dopamine agonists are diminished and such medications are no longer suitable” – even when advanced PD patients develop motor fluctuations, these medications are still essential for symptom relief and while doses/timing/formulations may need to be adjusted, movement disorders specialists do not typically stop dopaminergic medications at this stage of the disease as this sentence implies. Another example: “Outside of this window, or when the prescribed dosage no longer has an optimal effect, patients may exhibit motor fluctuations and dyskinesias.” Most dyskinesias occur as a peak-dose effect, and not during "off" time; this sentence implies the opposite.

3. The sentence that starts at line 50, “This window refers to the period of time at which medication is most effective at reducing motor related [7]” is not grammatically correct (seems to be missing a word/words at the end?).

4. The sentence starting at line 57, “Therefore, the next step for a subset of patients who no longer appropriately respond to pharmacological intervention is neurosurgery by means of deep brain stimulation (DBS) of the subthalamic nucleus (STN)” also need some revision. Technically, robust response to dopaminergic medications during "on" time is considered a good predictor of response to DBS surgery; the current wording suggests that patients who fail to respond to PD medications should be referred for surgery which is not exactly accurate.

5. In the sentence starting on line 64 “In PD, dopaminergic degradation of the substantia nigra (SN) is thought to result in inhibition of direct pathways, and disinhibition of indirect and hyperdirect pathways which collectively lead to  the functional disinhibition of output to motor related areas of the cortex and consequently, uncontrolled movement [16]” – this should be reworded; the pathophysiology of PD and its effects on the connections within the basal ganglia are complex, but it is generally accepted that the primary problem in PD is not “uncontrolled movement” but too little movement (PD is considered a hypokinetic movement disorder).

6. While the STN may be the most common target for DBS in PD, the authors should consider acknowledging in the paper that it is not the only one. (For example, some centers prefer GPi, and most agree that it is the best target for particular indications; some PD patients may have DBS targeting VIM for recalcitrant tremor; and newer targets are also under investigation for particular symptoms).

7. In the section starting at line 83, it is unclear whether the “indirect techniques” mentioned by the authors also utilize neuroimaging (the first paragraph implies that the STN is targeted without patient-specific imaging, but the second paragraph mentions the use of CT and MRI).

8. The paragraph starting at line 116 is very helpful at clarifying the organization of the paper.

9. Reading this paper from the perspective of a Movement Disorders neurologist, some of the details re: particular imaging techniques are a bit beyond my scope (as they may be for other readers, as well). It may be helpful to clarify in more simple terms the implications of these findings for routine clinical practice (ideally in each section), where clinicians may not have access to the ideal pre-operative planning techniques described in the paper.

Author Response

Please see the PDF attached.

Round 2

Reviewer 3 Report

Dear Authors,

Thank you for your thoughtful responses to my last review. All of my major concerns have been addressed. Best of luck with your manuscript.

This manuscript is a resubmission of an earlier submission. The following is a list of the peer review reports and author responses from that submission.

Round 1

Reviewer 1 Report

Isaacs and colleagues have put together an in-depth look at imaging techniques and considerations for mapping the STN and surrounding structures in preparation for DBS surgery for Parkinson's Disease.

Major Recommendations:

I think overall the paper is very thorough and raises many good points about the limitations of current imaging techniques used for lead placement in DBS surgery. However, at times I found the organization of the paper difficult to follow. Perhaps it would be helpful to reorganize as follows:  first describe the procedures different centers use to plan for DBS pre-operatively, with discussion of the challenges faced by clinicians using standard clinical imaging techniques. Then, consider a discussion of the intra-operative techniques used to confirm lead placement, followed by mention of how the system is programmed post-operatively. Next, include the discussion of techniques used in research that may help optimize lead placement and prevent the need for lead replacement or removal going forward. Finally, discuss how some of these research techniques can be incorporated into clinical practice. 

The paper can also benefit from clarification re: which imaging techniques refer to the process at the authors' site vs. reported by others, keeping in mind that many of the processes described in the article are not standardized across centers that perform DBS surgery. 

Minor Recommendations:
1. Introduction - The authors discuss in the first paragraph that as Parkinson's disease progresses, the "therapeutic window beings to narrow." This section needs to be clarified, with a mention and explanations of motor fluctuations and how these are improved with DBS.  It may also be helpful to mention, for those who are not specialists in the field, what are the clinical goals that neurologists and patients hope to achieve when pursuing DBS surgery.

2. The authors also mention that patients may "incorporate stronger dopamine agonists into their regimen" - this phrase may benefit from rewording as I believe most movement disorders specialists would not agree that dopamine agonists are stronger or more effective than levodopa in managing symptoms of advanced Parkinson's disease.

3. Later in the introduction the authors mention " between 2013 and 2017, there were 711 bilateral DBS placement surgeries...of those 169 had to be replaced or removed."  It should be clarified if those number refer to leads or patients.

4. Under section 4, the authors should clarify if they are discussing the process that happens at their site (vs others), or mention that processes vary between sites (for example, not all centers have the ability to obtain intraoperative neuroimaging, and not all do MER).

5. Line 197: Needs clarification, since many sites use BOTH MER and MRI for STN identification in PD (first mapping with MRI pre-operatively, then MER for confirmation)

6. Line 205: Clarify that the UPDRS III is one of several rating scales that is used (for example, others prefer the MDS-UPDRS), and the axial symptoms mentioned actually are accounted for in these rating scales 

7. Line 667: Consider rewording - the fact that a patient has a poor outcome after DBS may be related to suboptimal lead placement but there are other reasons 

Author Response

Faculty of Social and Behavioral Sciences

Integrative Model-based Cognitive

Neuroscience (IMCN) Research Unit

University of Amsterdam

Nieuwe Achtergracht 129B

Postbus 159261018

Tel: 0622370061

Date: 20.06.20

Dear Reviewer 1,

We would like to thank you for giving us the opportunity to improve and resubmit our manuscript entitled ‘Methodological considerations for neuroimaging in Deep Brain Stimulation of the Subthalamic Nucleus in Parkinson’s Disease patients’ [jcm-830941] to the Journal of Clinical Medicine. We also thank you and the other reviewers for providing constructive feedback which we have now incorporated to improve the quality of our paper.

The following section includes a point-by-point reply to the comments of reviewer 1. The main comments are numbered and marked in bold and followed by our response in standard text, and changes made in the manuscript are in italics. Attached with the application are also a manuscript including highlights of changes made, and an unmarked version of the revised manuscript. Reference styles have been adapted where appropriate.

Again, we hope that we have adequately addressed all concerns raised by the reviewers and we look forward to your response. Please do not hesitate to contact us in case you require further information.

Very kind regards, on behalf of my co-authors,

Bethany Isaacs

Major Recommendations

  1. However, at times I found the organization of the paper difficult to follow. Perhaps it would be helpful to reorganize as follows:  first describe the procedures different centers use to plan for DBS pre-operatively, with discussion of the challenges faced by clinicians using standard clinical imaging techniques. Then, consider a discussion of the intra-operative techniques used to confirm lead placement, followed by mention of how the system is programmed post-operatively. Next, include the discussion of techniques used in research that may help optimize lead placement and prevent the need for lead replacement or removal going forward. Finally, discuss how some of these research techniques can be incorporated into clinical practice.

We agree that the paper is at times lengthy which can make it hard to follow. There exists a multifaceted nature of the limitations inherent to neuroimaging. Therefore, we decided to split up the paper to more general topics, such as voxel size or specific absorption rates, and introduce the specific limitations and issues that were central to these aspects of MRI. Unfortunately, having an entire section that states the limitations of MRI would be quite large and perhaps overwhelming in itself. We tried our best to ease our readers in to this topic, with the use of MRI in STN DBS being central to section 2, and includes a discussion on both direct and indirect methods of targeting that are integral to all DBS centers. And the discussion of MRI-specific limitations begins with field strength in section three (arguably the most important limitation) and so on. We have however amended the following in the end of the second section on line 117, page 3:

Therefore, the goal of this paper is to explain the current procedures for structural target identification of the STN for DBS in PD using MRI. Further, we aim to identify limitations that may contribute to suboptimal identification of the STN, and provide alternatives for optimizing MRI in order to visualise the STN. The organisation of topics is as follows: Field strength; Current procedures for intra and post-operative verification with microelectrode recordings; SAR limitations; Shimming and magnetic field corrections; Sequence types and contrasts; Voxel sizes; Motion correction; Registration and image fusion; Quantitative maps; Complications unrelated to pre-operative planning; and Conclusions. The suggestions are presented with the underlying expectation that more accurate visualisation can translate into more accurate targeting.

Moreover, an in-depth discussion on post-operative care and optimization of stimulation parameters was not covered as it does not innately rely on neuroimaging and beyond the scope of the current review. For clarification we have added a brief text with references on in section 4. Current procedures for intra and post-operative verification with microelectrode recordings on line 227, page 6:

More in-depth literature on practices for post-operative verification, stimulation programming and care can be found in [92–94] and references therein.

Similarly, we have added the following section within the conclusion, which provides some suggestions on how to incorporate research techniques in to clinical practice. This has been added in line 725, page 16:

Multi-site clinical trials can facilitate the optimization and validation of certain sequences. Sequences with identical parameters should be compared on identical MRI systems and different sites to ensure harmonization and reliability of MRI systems as well as validate the sequence itself. Additionally, different MRI system specific sequences should be compared against any potential deviations in planned target and implanted location as well as the clinical outcome of the operated patient. Similarly, access agreements to work-in-progress protocols from the MRI vendors would facilitate the development and optimization of sequences, as would open access to underling algorithms and adjustable parameters within to pre-operative planning software vendors (e.g., Medtronic, St. Judes, Brainlab, Abott, Nextim, Boston Scientific).

  1. The paper can also benefit from clarification re: which imaging techniques refer to the process at the authors' site vs. reported by others, keeping in mind that many of the processes described in the article are not standardized across centers that perform DBS surgery. 

We thank the reviewer for this comment. We have added the following explanation for the 3T and 7T imaging protocols used at our site within the text on line 400, page 9:

We attempted to use a T2* based UHF-MRI with a GRE-ASPIRE sequence [152] on a 7T Siemens MAGNETOM system (Siemens Healthcare, Erlangen, Germany) for STN DBS planning in PD patients. When overlaying clinical anisotropic (0.45x0.45x2mm) Turbo Shot Echo (TSE) whole brain 3T T2w sequence obtained with a Siemens 3T Prisma system, and sub millimetre resolution (0.5mm) isotropic 7T T2* partial volume covering the subcortex, the STN appeared elongated along the posterior direction on 7T.

Additionally, we have added a new paragraph to highlight the issue of site-specific sequences, comparisons and interpretation within the text on line 420, page 9:

It is important to note that the sequences described in this specific instance are not standardized across centers, and scanner vendors, field strengths, contrasts, and sequence parameters even within the same sequence type will differ across DBS centers and research institutes. This makes a direct comparison across the quality and replicability of MRI scans very difficult, and unless systems are harmonized, interpretations should be site specific. See [87, 131] for a comprehensive review on sequences used for imaging the STN.

Minor Recommendations:

  1. Introduction - The authors discuss in the first paragraph that as Parkinson's disease progresses, the "therapeutic window beings to narrow." This section needs to be clarified, with a mention and explanations of motor fluctuations and how these are improved with DBS.  It may also be helpful to mention, for those who are not specialists in the field, what are the clinical goals that neurologists and patients hope to achieve when pursuing DBS surgery.

We have added the following text starting on line 45, page 2, which we hope adequately explains the concept of a therapeutic window:

As a result, common and initially beneficial drug treatments become less effective in about 40% of patients after four to six years, at which point the disease progresses and the therapeutic window, which refers to the period of time at which medication is most effective at reducing motor related symptoms without causing side effects, begins to narrow [7]. Outside of this window, or when the prescribed dosage no longer has an optimal effect, patients may exhibit motor fluctuations and dyskinesias. Further, the severity and frequency of these fluctuations and dyskinesias will increase as the disease naturally advances over time. 

Additionally, the section within the next paragraph (line 58, page 2) beginning ‘The next step for a subset of patients who no longer appropriately respond to pharmacological intervention is neurosurgery by means of deep brain stimulation (DBS) of the subthalamic nucleus (STN) [10]’… explains the goals and assumed mechanisms of DBS of the STN. However, we hope that the extract ‘…which medication is most effective at reducing motor related symptoms without causing side effects…’ within the new text (noted above) clarifies this in advance.

  1. The authors also mention that patients may "incorporate stronger dopamine agonists into their regimen" - this phrase may benefit from rewording as I believe most movement disorders specialists would not agree that dopamine agonists are stronger or more effective than levodopa in managing symptoms of advanced Parkinson's disease.

Thank you for pointing this out. We have amended our statement on line 52, page 2 to the following:

At this stage, the reliability, consistency and durability of the response to levodopa or dopamine agonists are diminished and therefore such medications are no longer suitable. Increasing the dosages is not always feasible and alternative treatments such as device-aided therapies must be considered. Further, drug therapy in PD is associated with side effects that include nausea and vomiting, sleep disorders, hallucinations and delusions, and chronic treatment is associated with the aforementioned motor complications including fluctuations and dyskinesia [8,9].

  1. Later in the introduction the authors mention " between 2013 and 2017, there were 711 bilateral DBS placement surgeries...of those 169 had to be replaced or removed."  It should be clarified if those number refer to leads or patients.

We have amended our statement on line 80, page 2 to the following:

Of those 711 surgeries, 169 patients required the DBS system to be either replaced or removed entirely [29].

  1. Under section 4, the authors should clarify if they are discussing the process that happens at their site (vs others), or mention that processes vary between sites (for example, not all centers have the ability to obtain intraoperative neuroimaging, and not all do MER).

We have added the following text starting on line 194, page 5, which we hope adequately clarifies the differing pre-operative processes across different DBS sites:

The current standard practice within the Netherlands includes both pre-operative planning with neuroimaging methods, and intra-operative verification with MER.

And on line 206, page 5, we have added:

Not all centres use pre-operative CT or MRI and instead rely on standard coordinates with MER verification (and vice versa).

  1. Line 197: Needs clarification, since many sites use BOTH MER and MRI for STN identification in PD (first mapping with MRI pre-operatively, then MER for confirmation)

Please refer to the response of point 4 above. We hope this amendment clarified both points 4 and 5.

  1. Line 205: Clarify that the UPDRS III is one of several rating scales that is used (for example, others prefer the MDS-UPDRS), and the axial symptoms mentioned actually are accounted for in these rating scales 

Thank you for your input, we have amended our statement on line 221, page 5, to the following:

Axial motor symptoms such as bradykinesia, rigidity, stability, gait, posture and dysarthria are assessed with rating scales such as the UPDRS III or the MDS-UPDRS [89,90].

  1. Line 667: Consider rewording - the fact that a patient has a poor outcome after DBS may be related to suboptimal lead placement but there are other reasons 

We have amended our statement, now on line 685, page 14 to the following:

T2 relaxometry has been shown to predict motor outcome in some PD patients with STN DBS, where patients who have low T2 values may fail to show a clinical benefit [236].

Additionally, we have added the following section on line 692, page 15

  1. Complications unrelated to pre-operative planning

Lastly, we would like to mention that while this paper specifically refers to suboptimal placement of DBS leads due to the limitations of neuroimaging, negative outcomes of DBS application can arise independently of planning procedures and surgical expertise. For example, neurosurgery have been linked to brain deformation and shift, changes in cerebral spinal fluid volume and intracranial pressure, which may induce spatial variability both during the surgery and cause a shift in the implanted lead location during recovery [27, 237]. Similarly, DBS surgeries are associated with infection, (mostly found in the chest and connector) [238], reactive gliosis and gliotic scarring [239], hemorrhage either during the surgery or delayed (in less than 5%) [240], and although rare; cerebral pneumocephalus [241]. In all these cases, the DBS system may require reimplantation, replacement or removal.

Reviewer 2 Report

The manuscript by Isaacs et al presents a very comprehensive review of neuroimaging modalities, future possibilities and their potential role in the localisation of the STN in DBS surgery. The paper is well written, there are no language issues and the structure is accessible and well proportioned. the principles involved are well explained for the uninitiated in the physics of brain imaging.

The weight of reference support is considerable and appropriate. Some sections could perhaps be shortened and more succinct but the nature of the subject matter does benefit from the detail provided. 

The review is timely and topical given the general move away from MER towards image guided and verified approach in many new centres. The authors are realistic in relation to the practical limitations around access to UHF MRI such as licencing, availability and time requirements in an era of increasing demand on MRI access.

Very minor changes suggested, not directly affecting the scientific rigor of the manuscript:

Lines 45 - 48; levodopa every hour very unusual and notion of stronger dopamine agonists is not in keeping with clinical practice. More correct to say that the reliability, consistency and durability of the levodopa response is lost leading to consideration of advanced device-aided therapies  

Equally in line 49; drug therapy does not lead to worsening of Parkinson symptoms (bradykinesia, rigidity, tremor) but chronic treatment is associated with motor complications such as treatment related fluctuations and dyskinesia.

Line 65; the suggested increased risk of suicide that has been reported in some reports and series has not been definitively established so perhaps the authors should be less emphatic here. Reasonable however to cite the potential for neuropsychiatric complications with lead placement in the medial STN which is of relevance to the manuscript.

Line 191; missing parenthesis and a more up to date reference with long term data on an MRI guided and verified approach might be more suitable [e.g. Aviles-Olmos et al JNNP 2014 85(12)]

Brief passing reference could be made in the discussion to other less controllable factors that can contribute to imprecision in lead localisation such as brain shift, oedema and CSF leaks that may confound any advances in STN targeting. 

Author Response

Faculty of Social and Behavioral Sciences

Integrative Model-based Cognitive

Neuroscience (IMCN) Research Unit

University of Amsterdam

Nieuwe Achtergracht 129B

Postbus 159261018

Tel: 0622370061

Date: 20.06.20

Dear Reviewer 2,

We would like to thank you for giving us the opportunity to improve and resubmit our manuscript entitled ‘Methodological considerations for neuroimaging in Deep Brain Stimulation of the Subthalamic Nucleus in Parkinson’s Disease patients’ [jcm-830941] to the Journal of Clinical Medicine. We also thank you and the other reviewers for providing constructive feedback which we have now incorporated to improve the quality of our paper.

The following section includes a point-by-point reply to the comments of reviewer 2. The main comments are numbered and marked in bold and followed by our response in standard text, and changes made in the manuscript are in italics. Attached with the application are also a manuscript including highlights of changes made, and an unmarked version of the revised manuscript. Reference styles have been adapted where appropriate.

Again, we hope that we have adequately addressed all concerns raised by the reviewers and we look forward to your response. Please do not hesitate to contact us in case you require further information.

Very kind regards, on behalf of my co-authors,

Bethany Isaacs

Minor Recommendations:

  1. Lines 45 - 48; levodopa every hour very unusual and notion of stronger dopamine agonists is not in keeping with clinical practice. More correct to say that the reliability, consistency and durability of the levodopa response is lost leading to consideration of advanced device-aided therapies  

Thank you for pointing this out. We have amended our statement , now bon line 52, page 2 to the following:

At this stage, the reliability, consistency and durability of the response to levodopa or dopamine agonists are diminished and therefore such medications are no longer suitable. Increasing the dosages is not always feasible and alternative treatments such as device-aided therapies must be considered.

  1. Equally in line 49; drug therapy does not lead to worsening of Parkinson symptoms (bradykinesia, rigidity, tremor) but chronic treatment is associated with motor complications such as treatment related fluctuations and dyskinesia.

Again, we thank you for clarifying this point. We have added the following text, now starting on line 55, page 2:

Further, drug therapy in PD is associated with side effects that include nausea and vomiting, sleep disorders, hallucinations and delusions, and chronic treatment is associated with the aforementioned motor complications including fluctuations and dyskinesia [8,9].

  1. Line 65; the suggested increased risk of suicide that has been reported in some reports and series has not been definitively established so perhaps the authors should be less emphatic here. Reasonable however to cite the potential for neuropsychiatric complications with lead placement in the medial STN which is of relevance to the manuscript.

We have removed the risk of suicide from the list of possible neuropsychiatric side effects, now on line 70, page 3, and the sentence now reads:

While DBS may ameliorate between 60 to 90% of the motor related symptoms of PD, it can produce neuropsychiatric side effects and emotional or associative disturbances, with side effects ranging from hypomania, apathy, hallucinations and reckless behaviour, changes in moral competence and personality [20–23].

  1. Line 191; missing parenthesis and a more up to date reference with long term data on an MRI guided and verified approach might be more suitable [e.g. Aviles-Olmos et al JNNP 2014 85(12)]

I’m very sorry but I could not identify any missing parenthesis and therefore could not amend it. We have added the refence (Aviles-Olmos, I., Kefalopoulou, Z., Tripoliti, E., Candelario, J., Akram, H., Martinez-Torres, I., ... & Limousin, P. (2014). Long-term outcome of subthalamic nucleus deep brain stimulation for Parkinson's disease using an MRI-guided and MRI-verified approach. Journal of Neurology, Neurosurgery & Psychiatry85(12), 1419-1425.) to what was line 191 and is now line 203, page 5:

Once the target has been verified via intra operative neuroimaging, the leads will be permanently implanted and then connected to a cortical grid and a stimulator will be inserted under the chest [77–80].

  1. Brief passing reference could be made in the discussion to other less controllable factors that can contribute to imprecision in lead localisation such as brain shift, oedema and CSF leaks that may confound any advances in STN targeting. 

This is indeed an important topic to address. We have added the following section, starting line 692, page 15:

  1. Complications unrelated to pre-operative planning

Lastly, we would like to mention that while this paper specifically refers to suboptimal placement of DBS leads due to the limitations of neuroimaging, negative outcomes of DBS application can arise independently of planning procedures and surgical expertise. For example, neurosurgery have been linked to brain deformation and shift, changes in cerebral spinal fluid volume and intracranial pressure, which may induce spatial variability both during the surgery and cause a shift in the implanted lead location during recovery [27, 237]. Similarly, DBS surgeries are associated with infection, (mostly found in the chest and connector) [238], reactive gliosis and gliotic scarring [239], hemorrhage either during the surgery or delayed (in less than 5%) [240], and although rare; cerebral pneumocephalus [241]. In all these cases, the DBS system may require reimplantation, replacement or removal.

Reviewer 3 Report

The revised manuscript gives a comprehensive methodological overview om the possibilities and limitations in MR investigation of the subthalamic nucleus in relation to deep brain stimulation in Parkinson’s disease. Special attention is directed toward the use of ultrahigh field MR at 7 Tesla compared to imaging at conventional field strength, 1.5 and 3 Tesla.

The review deals with relevant aspect in relation to the subthalamic nucleus and deep brain stimulation. The referee has only minor suggestions.

Minor suggestions:

Line 878 ”The benefit of SWI in visualizing the STN for DBS appears to be field dependent, with no observable increase at 1.5T compared to 3T”. Does this imply that the findings at 1,5 were expected to be better?

Text to figure 2, line 3. Add ”at 3T”

Text to figure 3: Voxel sizes should be given for all conditions, also in the legend to the figure. The description in the text is clear for T2w but not for the T2's.